# Sequencing, Analysis and Organization of the Complete Genome of a Novel Baculovirus *Calliteara abietis* Nucleopolyhedrovirus (CaabNPV)

**DOI:** 10.3390/v16020252

**Published:** 2024-02-04

**Authors:** Wenyi Jin, Mijidsuren Byambasuren, Uranbileg Ganbold, Huixian Shi, Hongbin Liang, Miaomiao Li, Hongtuo Wang, Qilian Qin, Huan Zhang

**Affiliations:** 1State Key Laboratory of Integrated Management of Pests and Rodents, Institute of Zoology, Chinese Academy of Sciences, Beijing 100101, China; jinwenyi23@ioz.ac.cn (W.J.); uranbileg@ioz.ac.cn (U.G.); s18348965602@163.com (H.S.); wanght@ioz.ac.cn (H.W.); qinql@ioz.ac.cn (Q.Q.); 2University of Chinese Academy of Sciences, Beijing 101408, China; 3Institute of Plant Protection, Mongolian University of Life Science, Ulaanbaatar 627153, Mongolia; bybyambasuren4747@gmail.com; 4Key Laboratory of Zoological Systematics and Evolution, Institute of Zoology, Chinese Academy of Sciences, Beijing 100101, China; lianghb@ioz.ac.cn; 5Institute of College of Basic Medicine, Shaanxi University of Chinese Medicine, Xianyang 712046, China; 1351029@sntcm.edu.cn

**Keywords:** *Alphabaculovirus*, baculovirus genome, *Calliteara abietis*

## Abstract

*Baculoviridae*, a virus family characterized by a single large double stranded DNA, encompasses the majority of viral bioinsecticides, representing a highly promising and environmentally friendly pesticide approach to insect control. This study focuses on the characterization of a baculovirus isolated from larvae of *Calliteara abietis* (Erebidae, Lymantriidae) collected in Mongolian pinaceae forests. This new isolate was called *Calliteara abietis* nucleopolyhedrovirus (CaabNPV). CaabNPV exhibits an irregular polyhedron shape, and significant variation in the diameter of its occlusion bodies (OBs) was observed. Nucleotide distance calculations confirmed CaabNPV as a novel baculovirus. The CaabNPV genome spans 177,161 bp with a G+C content of 45.12% and harbors 150 potential open reading frames (ORFs), including 38 core genes. A comprehensive genomic analysis categorizes CaabNPV within Group II alphabaculovirus, revealing a close phylogenetic relationship with *Alphabaculovirus orleucostigmae* (OrleNPV). Additionally, repeat sequence analysis identified three highly repetitive sequences consisting of 112 bp repeat units, known as homologous regions (*hrs*). This research contributes valuable insights into CaabNPV’s phylogenetic placement, genomic structure, and its potential applications in insect biocontrol.

## 1. Introduction

In the current context, approximately one hundred species of baculoviruses have been identified, with a focus on harnessing their insecticidal potential for protecting field crops [1]. However, the prevailing challenge lies in the host specificity exhibited by the majority of these baculoviruses, underscoring the urgent need for ongoing exploration and the development of novel viral strains. Simultaneously, the discovery of new baculovirus holds paramount significance, not only for advancing crop protection strategies but also for garnering essential insights into their evolution and intricate interactions with host organisms [2]. This exploration not only enhances the arsenal of crop protection methodologies but also contributes significantly to our understanding of the dynamic interplay between viruses and their hosts.

*Baculoviridae*, a family of large double-stranded DNA viruses, emerges as a promising and environmentally friendly pesticide with the capability to address the threats posed by insects across various orders, including Lepidoptera, Hymenoptera, Diptera, and others [3]. Characterized by a single circular double-stranded DNA ranging from 80 to 180 kb [4] and encoding 80 to 190 genes [4], the genome is enclosed within rod-shaped nucleocapsids measuring 230 to 385 nm in length and 40 to 60 nm in diameter [5]. The distinctive life cycle of *Baculoviridae* involves the production of two progeny viruses with divergent structures and functions. One viral phenotype, referred to as budded virus (BV), facilitates systematic infection spread between cells within a host, while the other, known as occlusion-derived virus (ODV), transmits infection from insect to insect via the oral route [6]. This ancient virus demonstrates the capacity for co-evolution with its original host [7,8]. One defining characteristic of baculoviruses is the presence of a conserved set of 38 genes which form a shared core that spans their entire genomic structure. The classification of baculoviruses into four distinct genera: *Alphabaculovirus* (lepidopteran-specific nucleopolyhedroviruses [NPVs]), *Betabaculovirus* (lepidopteran-specific granuloviruses [GVs]), *Gammabaculovirus* (hymenopteran-specific NPVs), and *Deltabaculovirus* (dipteran-specific NPVs), respectively. Additionally, *Alphabaculovirus* further divides into Group I and Group II [4]. Group I alphabaculoviruses utilize GP64 as their envelope fusion protein (EFP) in the BV, while Group II alphabaculoviruses lack GP64 and instead employ F protein as their EFP [9].

*Calliteara abietis* (Erebidae: Lymantriidae), a significant pest of Pinaceae, is prevalent in northern and eastern Europe, but has limited presence in northern Asia. Targeting primarily *Pinus sylvestris*, larch trees, and fir trees, this pest causes extensive damage to pine forests across vast territories [10]. Its wide distribution and destructive feeding habits underscore its impact on forestry ecosystems in these regions.

Genome research serves as a crucial source of illumination, unveiling the enigmatic facets of baculoviruses. Through meticulous analysis, baculoviruses can be precisely identified and classified, revealing the intricacies of their functional genes and encoded proteins. Beyond providing new theoretical foundations for virus control, this study initiates a comprehensive exploration into the biology of the virus and its intricate interactions with host organisms. In doing so, it establishes a robust foundation that not only deepens our current understanding but also serves as a springboard for future research endeavors. However, given the narrow host specificity of most baculoviruses, the discovery and sequence analysis of new baculoviruses remain crucial.

In this study, the Erebidae larvae exhibiting signs of baculoviruses infection were collected from a forest in Mongolia. Through a combination of morphological characteristics and mitochondrial cytochrome oxidase I (COI) sequence data, the host was identified as *C. abietis*. The baculovirus isolated from the larvae underwent a complete genome analysis, designated as *Calliteara abietis* nucleopolyhedrovirus (CaabNPV). Comparative analysis with other fully sequenced baculovirus genomes confirmed that CaabNPV represents a novel virus.

## 2. Materials and Methods

### 2.1. Baculovirus Isolation and Host Identification

CaabNPV and its host larva were collected from Bayan-Adraga, Mongolia (coordinates: 48°32′14.5″ N, 111°04′44.5″ E, altitude 1034 m) in 2023. *C. abietis* larvae were collected, including uninfected individuals and individuals showing signs of infection.

Insect DNA samples were extracted using the MightyPrep Reagent for DNA (TAKARA). The amplification of the COI barcode was conducted using primers specifically designed for the Erebidae [11]. The amplified fragments were subjected to Sanger sequencing, and the obtained sequences were compared and analyzed against the barcode sequences available in GenBank [12]. Primer sequence information can be found in Appendix A.

The OBs were derived from the larvae cadavers that succumbed to the infection, and purification of occlusion bodies (OBs) was performed through a series of steps, including maceration, homogenization, filtration, centrifugation, and sucrose density-gradient ultracentrifugation [13]. 

### 2.2. Electron Microscopy

Baculovirus-specific characteristics were examined utilizing light microscopy (Leica DM2000, Leica, Wetzlar, Germany) to assess the purity of the baculovirus.

For a more in-depth exploration of structural nuances, purified OBs underwent detailed observation using a scanning electron microscope (SEM, Hitachi, Tokyo, Japan) with a 5 kV acceleration voltage. The dimensions of CaabNPV OBs were precisely ascertained, relying on measurements extracted from a comprehensive set of 69 complete SEM image samples.

### 2.3. Viral DNA Isolation, Sequencing, and Assembly

The purified OBs was resuspended in an alkaline solution (100 mM Na_2_CO_3_, pH = 10.8) and incubated at 37 °C for 30 min. The pH of the solution was adjusted to 8.0 using 1M hydrochloric acid. Ribonuclease A and Proteinase K was added to the solution and incubated, following the initial incubation at 37 °C for 2 hours, the solution was subjected to a subsequent incubation at 65 °C for an additional 2 hours. Viral DNA was extracted using a phenol/chloroform/isoamyl alcohol (25:24:1) mixture and ethanol-precipitated. The DNA concentration was determined using a spectrophotometer, and the purity was validated by the 260/280 ratio.

CaabNPV full genome sequencing was performed on DNB-seq. After data filtration, the average read length was determined to be 150 bp, and the achieved coverage reached 38,135×. The Q20 and Q30 contents were 96.14%, and 90.44%, respectively. Unicycler (version v0.5.0) was used to assemble high-quality pair-end reads into contigs by de novo assembly. 

The A region with a low mapping rate was amplified through PCR and assembly errors were resolved through Sanger sequencing [14]. Additionally, CaabNPV exhibits an assembly ambiguity region. To identify this region, vIdentify-F/R primers (Appendix A) were utilized for the amplification of both CaabNPV and *C. abietis*. To ensure that the experimental sample of *C. abietis* was not contaminated with baculovirus, specific primers lef8-F/R (Appendix A) were employed to amplify the baculovirus for host larvae identification.

The amplification process followed the manufacturer’s instructions using TransStart^®^ FastPfu DNA Polymerase (Beijing, TransGen Biotech Co., Ltd., Beijing, China). The reaction conditions were as follows: initial denaturation at 95 °C for 2 minutes, followed by 35 cycles of denaturation at 95 °C for 20 seconds, annealing at 55 °C for 20 seconds, and extension at 72 °C for 30 seconds. A final extension was performed at 72 °C for 5 minutes.

### 2.4. Genome Sequence Analysis

Open reading frames (ORFs) encoding proteins with a minimum of 50 amino acids were predicted using both FGENESV [15] and the NCBI ORF Finder. The annotation convention for overlapping ORFs, limited to 75 base pairs, involves annotating only the larger ORF, unless the smaller one corresponds to a conserved baculovirus homolog [16]. This approach stems from the assumption that the expression of the smaller ORF is less probable. ORFs within homologous region (*hr*) sequences, due to their relative instability, were left unannotated.

The identified ORFs were annotated based on homology using the Protein-Protein BLAST algorithm. The entire genomic data were submitted to Genbank under accession number PP171514.

Gene parity plots were generated to compare ORF organization between CaabNPV and the selected baculoviruses [17].

EMBOSS stretcher [18] was employed to assess the genomic homology between CaabNPV and other baculoviruses. *hrs* were identified using the tandem repeats finder [19], and EMBOSS palindrome [20] was utilized to search for palindrome sequences. The ViennaRNA secondary structure server [21] was applied to predict the secondary structure of the conserved *hr* sequence.

To identify early and late promoter motifs, a 180-nucleotide upstream region of each initiation codon was screened. The criteria for identifying the early promoter component included the presence of a common TATA-box motif (TATAW) with a CAKT mRNA start site sequence located 25–35 nucleotides downstream [22,23]. Moreover, elements similar to TATA, such as (TAATWAA), were identified upstream of early expression factors [24]. The late promoter component was defined as DTAAG, typically located around -60 nucleotides upstream of the ORF initiation codon [25].

### 2.5. Phylogeny and Kimura 2-Parameter Analysis

Protein sequences encoded by the 38 core genes were extracted from 108 baculovirus genomes, with inclusion of CaabNPV (Appendix A), and aligned using the MAFFT method with auto strategy and normal alignment mode [26]. The alignment was performed using OrthoFinder (v2.5.5) with default parameters [27]. Following sequence alignment, the genomes were concatenated while maintaining the order of each gene. The optimal partitioning scheme for maximum likelihood (ML) was determined using RAxML-NG (v1.2.0) [28]. The best partitioning scheme for ML was identified by PartitionFinder2 using the Bayesian information criterion (BIC) and a greedy search algorithm with branch lengths linked [29]. The ML tree was inferred with IQ-TREE under Ultrafast bootstrap with 5000 replicates.

Kimura 2-parameter (K2P) pairwise distances [30] from aligned nucleotide sequences of *polyhedrin/lef-8/lef-9* fragments were calculated separately using the pairwise distance calculation in PHYLIP (v 3.6). Substation rates among sites were set to be uniform. Gaps within the alignment were treated as missing data, ensuring a comprehensive analysis of sequence variations.

## 3. Results and Discussion

### 3.1. Host Determination and Virus Characterization

In Mongolia, viruses and their host larva were collected in the field (Figure 1a,b). Based on larval morphology, they were classified as a larvae species in Calliteara genus (Erebidae: Lymantriidae). The deceased larvae samples collected on pine trees exhibited pathological features associated with baculovirus infection (Figure 1b). These characteristics included body darkening, liquefaction, and hanging upside down on branches. Sequencing of the mitochondrial COI barcode (GenBank accession number: KX436516.1) confirmed the host as *C. abietis*. The sequencing results can be found in Appendix B. Consequently, the virus was named *Calliteara abietis* nucleopolyhedrovirus (CaabNPV).

Under optical microscopy, polyhedral of OBs displaying Brownian motion were observed. SEM analysis of CaabNPV unveiled an ultrastructure characterized by irregularly shaped polyhedra (Figure 1c), indicative of a typical feature observed in OBs of NPV. Quantification of individual OBs demonstrated a mean diameter of approximately 2.19 ± 0.56 μm, falling within the size range previously reported for baculoviruses in the literature [5]. The size variation observed in CaabNPV OBs may be attributed to its natural collection, indicating that the strain has not undergone purification to establish a uniform viral population. This observation aligns with established findings, further supporting the classification of CaabNPV within the *Alphabaculovirus* genus.

### 3.2. Nucleotide Sequence Analysis

Utilizing the DNB-seq high-throughput sequencing platform, a total of 4,023,799 high-quality pair-end reads were generated for the CaabNPV sample. The complete genome was assembled using Unicycler v0.5.0. Subsequently, validation of ambiguous regions was conducted through PCR and Sanger sequencing methods.

The analysis of the assembled results for the ambiguous region revealed the presence of two variant assemblies. As a result of the baculovirus being purified before sequencing, any potential host fragments were effectively removed, indicating that both identified segments originated from CaabNPV. The region is located downstream of the *polh* initiation site at 16,792 bp. One variant comprises a 189 bp repetitive sequence interspersed with a 793 bp contig, while the other variant consists of a single 189 bp contig (Figure 2a, with sequence details provided in Appendix C). PCR and Sanger sequencing verification using vIdentify-F/R primers further confirmed the simultaneous existence of these two variants within CaabNPV (Figure 2b). To eliminate the possibility of host contamination, samples were collected from wild host larvae without baculovirus infection. Verification using baculovirus gene primers lef8-F/R [31] confirmed the absence of CaabNPV infection in the host (Figure 2c lane 3, positive control in Figure 2c lane 4). Subsequent amplification of host DNA using vIdentify-F/R primers resulted in the generation of two expected bands, as depicted in Figure 2c lane 2. Further sequencing results indicated the presence of both the 189 bp and 1171 bp variants in the uninfected host genome, affirming the concurrent existence of these two variants within the host and CaabNPV genome.

The presence of a transposon-like structure is likely to be attributed to horizontal gene transfer from the host genome during virus replication [32,33]. Additionally, the coexistence of two distinct genomes within a host organism was observed. This phenomenon may be attributed to the natural survival of a mixed population, comprising multiple genotypes, of an insect nucleopolyhedrovirus. Such diversity is likely to be maintained to facilitate the efficient transmission of the virus by insects. Infections involving multiple genotypes tend to generate OBs with higher pathogenicity compared to those produced by any single genotype alone [34].

However, due to the inability to artificially rear host larvae and obtain purified individual baculoviral strains, the determination of whether the ambiguous sequence results from horizontal gene transfer between the baculovirus and host or is due to the inherent presence of multiple genotypes within OBs is hindered. Despite inferring based on contig depth, with a depth of 2.01× for the 189 bp contig and 0.98× for the 793 bp contig, a comprehensive understanding of the virus structure and function remains constrained by the lack of detailed explanations for such transposon-like structures.

Furthermore, contig depth-based estimation revealed that the chosen assembly result for further analysis of the CaabNPV genome includes two repetitive 189 bp sequences interspersed with a 793 bp sequence (Figure 2a). Due to the absence of homologous gene matches and unreported host genome sequences, the current elucidation of this transposon-like structure remains inadequate. This limitation will impede a comprehensive understanding of virus structure and function.

The circular genome of CaabNPV measured 177,161 bp in size. The CaabNPV genome exhibited a G+Cs content of 45.12%, similar to the average GCs content observed in group I (44.9%) and group II (41.6%) alphabaculoviruses [35]. Following the original definition, the polyhedrin gene was designated as ORF1, and its methionine start codon was defined as the zero point on the CaabNPV physical map (Figure 3).

In total, 150 putative ORFs were identified (Tabel S3), each with a minimum length of 50 amino acids. Remarkably, CaabNPV encoded a higher number of genes (150 ORFs) compared to other baculoviruses, where the average number of ORFs was 140.5 [36]. These ORFs collectively accounted for 83% of the entire genome. Among them, 83 (55.3%) ORFs were oriented in the forward direction, while 67 (44.6%) were in the reverse orientation (Figure 3). CaabNPV encompassed 38 core genes shared across all baculoviruses (Appendix A). It also contains 23 genes that were conserved among baculoviruses [37] and had 72 common genes found in the *Baculoviridae* family.

In the upstream region preceding the start codon of the 17 unique genes in CaabNPV, spanning 180 base pairs, a comprehensive analysis was conducted. Within the 3 ORFs (Caab_082, Caab_138, Caab_139), early promoter sequences were identified, comprising TATA box motifs (TATAW and TAATWAA) and CAKT. The average distance between these motifs was determined to be 29.6 bp. Furthermore, eight ORFs (Caab_020, Caab_028, Caab_036, Caab_080, Caab_081, Caab_099, Caab_100, Caab_136) contained the late promoter element (DTAAG), which was positioned at an average distance of -68 bp upstream of the start codon. Notably, two ORFs (Caab_087, Caab_137) exhibited both early and late promoter motifs. Interestingly, a total of four ORFs (Caab_027, Caab_101, Caab_116, Caab_117) lacked recognizable typical promoter motifs. Further investigations are essential to determine the functional significance of these findings in CaabNPV.

### 3.3. Phylogenetic Analysis

To investigate the evolutionary relationship between CaabNPV and other baculoviruses, a phylogenetic tree was constructed based on the amino acid sequences of 38 highly conserved core genes from 108 baculovirus genomes (Appendix A). CaabNPV encodes the F protein and lacks the *gp64* gene (Figure 3), placing it within the Group II of genus *Alphabaculovirus*. In the ML phylogram, alphabaculoviruses clustered into a monophyletic group, and Group I alphabaculoviruses and Group II alphabaculoviruses formed a monophyletic clade within the alphabaculovirus group. This topological structure is in accordance with previous studies [9]. Based on the phylogenetic tree analysis, CaabNPV is classified within the Group II alphabaculovirus (Figure 4). CaabNPV is a sister species to *Alphabaculovirus orleucostigmae* (OrleNPV) and has a nucleotide identity of 47.2%.

Based on the pairwise nucleotide distances estimated using the K2P model with *polyhedrin/granulin*, *lef-8*, and *lef-9* genes, whether CaabNPV was a novel baculovirus was identified [31]. The K2P pairwise for CaabNPV and other baculovirus above-mentioned genes were greater than 0.05 substitutions/site (Appendix A), indicated that CaabNPV was a new baculovirus.

The gene parity plot was used to compare the gene order among different baculovirus genomes. Its purpose was to illustrate the collinearity among baculovirus genomes and the patterns of baculovirus evolution [17]. Culex nigripalpus NPV (CuniNPV, *Deltabaculovirus*, NC_003084), Neodiprion sertifer NPV (NeseNPV, *Gammabaculovirus*, NC_005905), Adoxophyes orana granulovirus (AdorGV, *Betabaculovirus*, NC_005038), Autographa californica nucleopolyhedrovirus (AcMNPV, group I alphabaculovirus, NC_001623), Lymantria dispar MNPV (LdMNPV, group II alphabaculovirus, NC_001973), were used as representative baculoviruses, and OrleNPV (NC_010276), Hemileuca sp. nucleopolyhedrovirus (HespNPV, NC_021923) and Euproctis pseudoconspersa nucleopolyhedrovirus (EupsNPV, NC_012639) were used as related baculoviruses. The gene order of CaabNPV was compared with the genomes mentioned above. CaabNPV shared 36 homologous ORFs with CuniNPV, 49 with NeseNPV, 77 with AdorGV, 105 with AcMNPV, 120 with LdMNPV, 112 with OrleNPV, 112 with HespNPV and 112 with EupsNPV.

The whole-genome nucleotide identity between the CaabNPV genome and those of OrleNPV, HespNPV, and EupsNPV was 47.2%, 44.1% and 44.4%, respectively. CaabNPV exhibited a highly collinear gene order with OrleNPV and EupsNPV, indicating a similarity in gene structure. The gene parity plots reveal that the majority of ORFs in OrleNPV and EupsNPV are arranged in an inverse order compared to the CaabNPV ORFs. This is attributed to the artificial definition of *polh* as the first ORF, leading to the occurrence of an inversion in the region encompassing the *polh*. In contrast, its gene arrangement significantly differed from that of NeseNPV (*Gammabaculovirus*), CuniNPV (*Deltabaculovirus*), and AdorGV (*Betabaculovirus*) (Figure 5).

### 3.4. Repeated Sequences

Most of baculovirus *hrs* contain repeated AT-rich sequences [38]. The *hrs* are characterized by comprising a series of tandem repeats, which include an imperfect palindromic core, interspersed at different locations in the genome [16]. The *hrs* play putative or demonstrated roles as origins of DNA replication and enhancers of baculovirus early promoters [39].

The CaabNPV genome contained three *hrs*, and each consisted of between two and four repeat units of approximately 112 bps. Near the repeat unit center, there was an imperfect palindrome spanning 26 bps (Figure 6a,b). As depicted in Figure 3, *hr1* is located between Caab_27 and Caab_28, and *hr2* between Caab_37 and Caab_38; they contain four and three repeat units, respectively. *hr3* was positioned between Caab_118 and Caab_119, featuring a tandem repeat of two copies. Secondary structure predictions were illustrated in Figure 6c, and the *hr1* contained four repeating units, showing a more complex secondary structure.

### 3.5. Analysis of CaabNPV Multi-Copy Genes

The majority of sequenced baculovirus genomes exhibit a range of 1–16 copies of the *bro*, and its function is not clear yet [40]. The Bro protein possesses DNA-binding capabilities and is associated with the modulation of host DNA replication in *Alphabaculovirus bomori* (BmNPV) [41]. In CaabNPV, there were a total of 11 Bro proteins, among which multiplied Bros were adjacent (Caab_105- Caab_107 and Caab_111- Caab_114).

Except for CuniNPV, homologs of the DNA-binding protein (DBP) are evident in all cataloged baculovirus genomes. Multiple copies of DBP genes were observed in Group II alphabaculoviruses [42]. The DBP gene plays a role in the unwinding and annealing of DNA during the replication process [43]. Notably, the CaabNPV genome contained two copies of the DBP gene.

The P26 gene, a unique feature specific to baculoviruses and consistently positioned, is variably present in some baculoviruses, including instances of a second copy. CaabNPV, for example, two copies of the P26 gene are encoded, with each situated adjacent to P10 and IAP-3, mirroring the presence of two copies of P26 in EranNPV [42].

Except for *Alphabaculovirus urprotei* (UrprNPV), all *Alphabaculoviruses* possess the Chab protein, and typically have two copies [42]. However, in the CaabNPV genome, only a single copy of Chab was evident.

## 4. Conclusions

In this study, the host was identified as *C. abietis* through morphological examination and COI barcode sequencing, and we first identified and named a baculovirus that was isolated from *C. abietis* larvae collected in Mongolia as *Calliteara abietis* nucleopolyhedrovirus (CaabNPV). SEM analysis revealed the irregularly polyhedra ultrastructure of CaabNPV, consistent with features characteristic of aphabaculoviruses OBs.

Despite the limitation of lacking purified viral strains for detailed analysis of ambiguous regions, high-throughput sequencing and genome assembly provided valuable insights. Based on the estimated pairwise nucleotide distances, CaabNPV has been identified as a novel baculovirus. Phylogenetic analysis placed CaabNPV within the aphabaculoviruses Group II, with OrleNPV identified as its closest relative, showing a high degree of collinearity. Genomic structure analysis unveiled a higher number of ORFs in CaabNPV, some of which are unique compared to other baculoviruses.

In summary, the comprehensive analysis of CaabNPV genome sheds light on its biological features and molecular mechanisms, providing a foundation for research on the diversity, evolution, and host interactions of baculoviruses. Further investigations are warranted to elucidate the ecological roles of CaabNPV and the functional significance of its genomic characteristics.

## Figures and Tables

**Figure 1 viruses-16-00252-f001:**
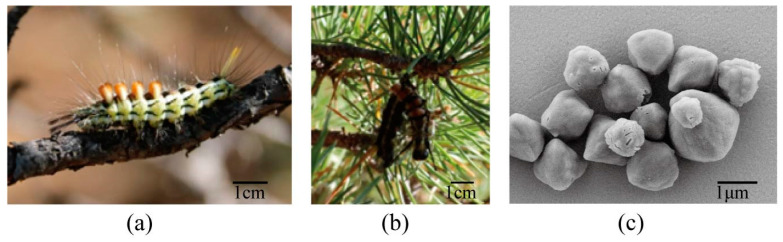
*Calliteara abietis* larva and *Calliteara abietis* nucleopolyhedrovirus (CaabNPV) scanning electron microscope (SEM). (**a**) A *C. abietis* larva feeding on pine tree. (**b**) Deceased *C. abietis* displaying pathological features indicative of baculovirus infection. (**c**) SEM picture depicting irregularly shaped CaabNPV occlusion bodies (OBs).

**Figure 2 viruses-16-00252-f002:**
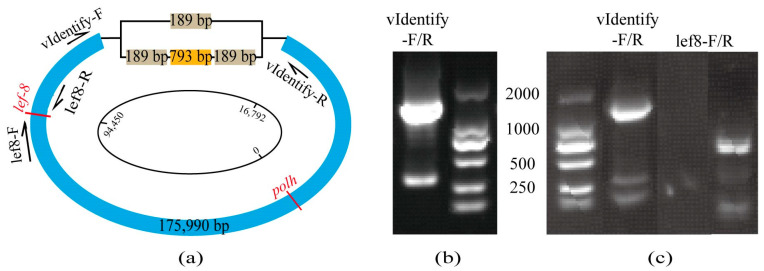
CaabNPV assembly results (**a**) The schematic diagram illustrates the assembly results of CaabNPV, emphasizing ambiguous regions. Arrows indicate primer sequences, and gene names are denoted in red, the light brown box represents a 189bp fragment, the orange box represents a 793bp fragment, and the remaining part of the CaabNPV genome is represented by the blue box. (**b**) PCR amplification of ambiguous regions in CaabNPV with vIdentify-F/R reveals large fragments consisting of two repeated 189 bp sequences interspersed with a 793 bp sequence, while small fragments consist of a singular 189 bp sequence. (**c**) Lane 2: Amplification of ambiguous regions in host larvae, depicting PCR amplification of ambiguous regions in the host larvae; Lane 3: Amplification of the viral *lef8* fragment in host larvae; Lane 4: *lef8* positive control with CaabNPV as the sample, resulting in an amplification size of 849 bp.

**Figure 3 viruses-16-00252-f003:**
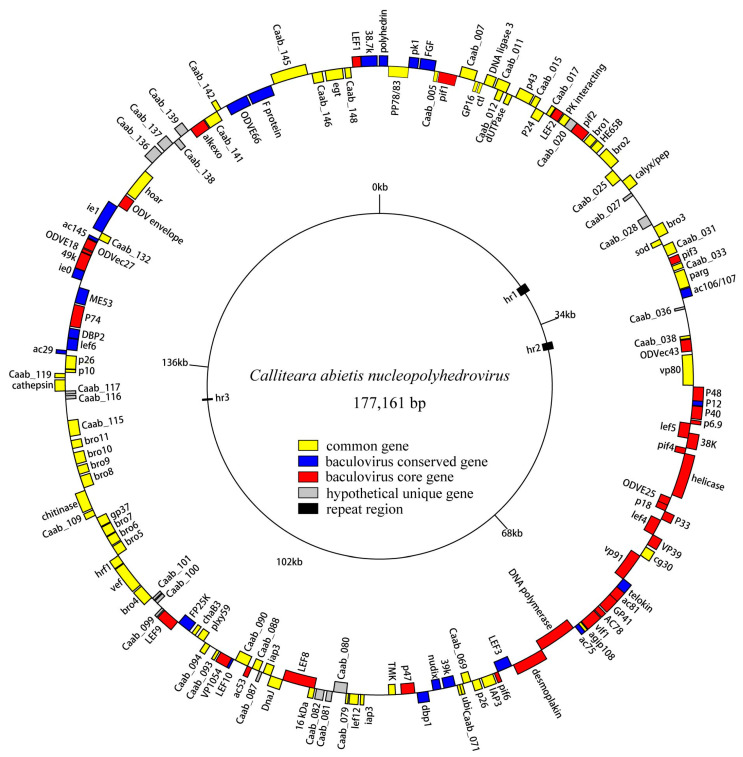
Circular map of CaabNPV genome. The colors represent gene types: red represents baculovirus core genes, blue represent baculoviruses conserved genes, yellow represents other baculovirus common genes, gray represents hypothetical genes unique to CaabNPV, black represents repeat regions.

**Figure 4 viruses-16-00252-f004:**
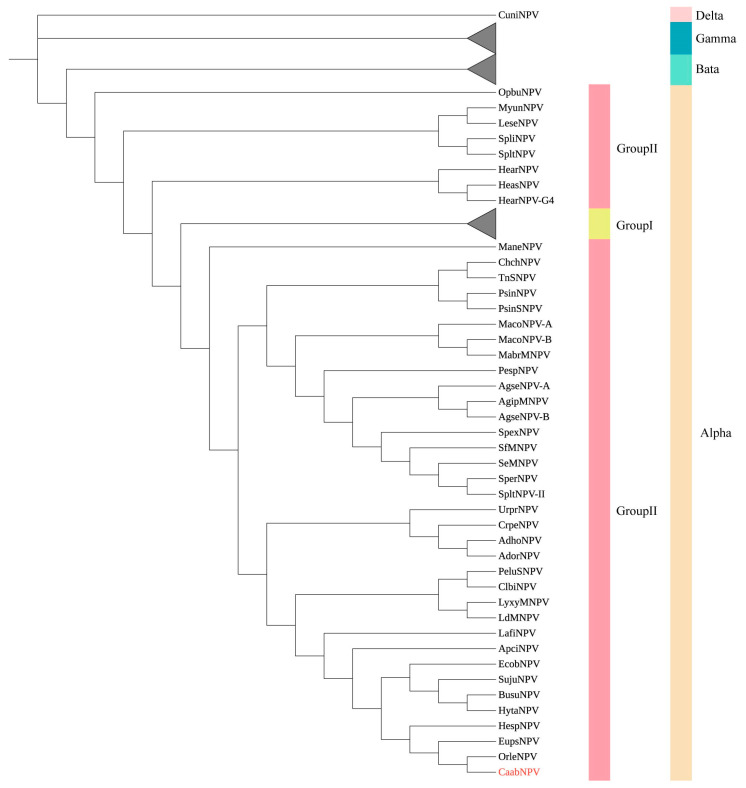
Unrooted phylogenetic tree created from the concatenated 38 core genes found in 108 sequenced baculovirus genomes. CaabNPV is marked in red. The tree was constructed using the minimum evolution method with 1000 bootstrap. Clusters representing the group I alphabaculovirus genera, *Beta-*, *Gammabaculovirus* have been simplified for clarity.

**Figure 5 viruses-16-00252-f005:**
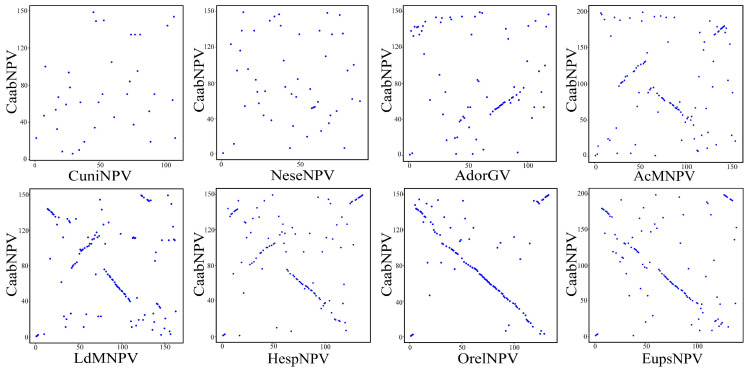
Gene parity plots comparing CaabNPV with other baculoviruses. The ORF content and order of the CaabNPV genome (y-axis) was compared with that of Culex nigripalpus NPV (CuniNPV, *Deltabaculovirus*), Neodiprion sertifer NPV (NeseNPV, *Gammabaculovirus*), Adoxophyes orana granulovirus (AdorGV, *Betabaculovirus*), Autographa californica nucleopolyhedrovirus (AcMNPV, group I alphabaculovirus), Lymantria dispar MNPV (LdMNPV, group II alphabaculovirus) and Orgyia leucostigma NPV (OrleNPV), Hemileuca sp. nucleopolyhedrovirus (HespNPV) and Euproctis pseudoconspersa nucleopolyhedrovirus (EupsNPV) (x-axes). Each point in the plot represented the relative position of open reading frames in the CaabNPV genome with those from other baculoviruses.

**Figure 6 viruses-16-00252-f006:**
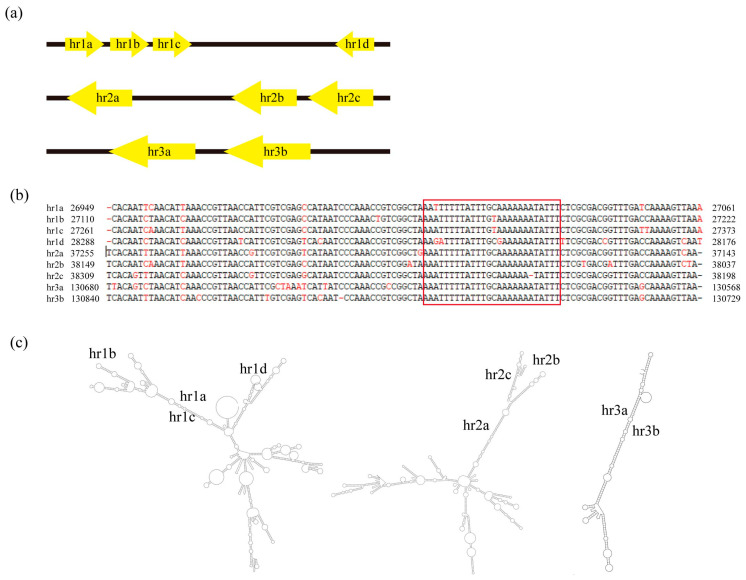
Analysis of CaabNPV homologous regions (*hrs*). (**a**) The schematic diagram for location and structure of the five *hrs* in the CaabNPV genome. (**b**) Sequence alignment of CaabNPV *hrs* repeat units. The imperfect palindrome sequence was represented with Group II alphabaculoviruses in the red box. (**c**) Secondary structure prediction of the conserved *hrs* sequence.

## Data Availability

The research data has been successfully uploaded to the open database Genbank to support manuscript submitted.

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
