# Peer review of "Sequencing, Analysis and Organization of the Complete Genome of a Novel Baculovirus *Calliteara abietis* Nucleopolyhedrovirus (CaabNPV)"

_viruses, 2024, doi:10.3390/v16020252_

Round 1
Reviewer 1 Report
Comments and Suggestions for Authors
In this manuscript, the authors report the isolation and genome sequence determination of a new alphabaculovirus from an erebid moth, Calliteara abietis. The authors identified and collected larvae exhibiting signs of nuclear polyhedrosis in a forest (presumably) in eastern Mongolia, extracted occlusion bodies from them, and sequenced the DNA from alkali-liberated occlusion-derived virus. ORFs and other features of the sequence (including an apparent transposon insertion) were characterized, as was the phylogenetic relationships between this virus (Callitear abietis nucleopolyhedrovirus, or CaabNPV) and other baculoviruses.
Major issues
1) Title, line 350, and throughout the manuscript: The authors initially refer to this virus as “Alphabaculovirus Caabietis (CaabNPV)”. “Alphabaculovirus Caabietis” (sic) is in a Linnaean binomial format that is reserved for species names, but this is a new virus and no species has been created for it by the International Committee on Taxonomy of Viruses (ICTV). Based on the naming convention followed by the baculovirus community and on the abbreviation the authors use (CaabNPV), the name of this virus should be “Calliteara abietis nucleopolyhedrovirus”. Instances of “Alphabaculovirus Caabietis” should be replaced with this name.
The authors confuse virus names and species names in this manner throughout the manuscript. For example, in lines 288-292, the authors write, “Deltabaculovirus cunigripalpi (CuniNPV), Gammabaculovirus nesertiferis (NeseNPV), Betabaculovirus adoranae (AdorGV), Al phabaculovirus aucalifonicae (AcMNPV, group I), Alphabaculovirus lydisparis (LdMNPV group II), were used as representative baculoviruses, and OrelNPV, Alphabaculovirus leucae (HespNPV) and Alphabaculovirus eupseudoconspersae (EupsNPV) were used as related baculoviruses.” The binomial names in italics are species names, not virus names. A species is not the same as a virus; a species “…is a monophyletic group of MGEs (mobile genetic elements) whose properties can be distinguished from those of other species by multiple criteria.” (ICTV code, section III, 3.20, https://ictv.global/about/code), Thus, in line 288, Deltabaculovirus cunigripalpi is a species, not a baculovirus. The authors should not use the names of species or other taxa to refer to viruses – they should use the names of the actual virus isolates whose sequences and other physical properties they are describing, using, or referring to.
The authors should consult Zerbini et al. 2022, https://doi.org/10.1007/s00705-021-05323-4, for information on how to write names of viruses, species, and other taxa. They should also consult Siddell et al. 2023 (DOI 10.1099/jgv.0.001840) for information on how the ICTV implements changes to official virus taxonomy.
2) The p6.9 ORF: The authors claim that there is no p6.9 ORF in the CaabNPV genome. This is hard to believe, given that it has been found in every other baculovirus genome sequenced to date next to lef-5. Even Neodiprion abietis nucleopolyhedrovirus has a short (21-aa) ORF next to lef-5 with the characteristic high % of arginine and serine residues. Some genomes do not have annotated p6.9 ORFs, likely because such ORFs often aren’t detected by database queries, but that doesn’t mean that p6.9 ORFs are not there. The authors should manually search for a p6.9 ORF next to the lef-5 ORF. If they still can’t find the p6.9 ORF, or if they find a version of p6.9 that appears to be truncated or missing a start codon, they should PCR-amplify the region where it should be and sequence the amplimer with dideoxy sequencing to confirm that p6.9 is really missing.
Along the same lines, the ICTV Online Report chapter on Baculoviridae has a Resources section with fasta files of individual core gene amino acid sequence alignments at https://ictv.global/report/chapter/baculoviridae/baculoviridae/resources. These alignments include sequences of core gene ORFs that were not annotated in the GenBank records of some genome sequences (including p6.9 ORFs). Though it may not make that much of a difference to the topology of the Figure 4 tree or to the authors’ conclusions about the relationships of CaabNPV to other alphabaculoviruses, the authors may wish to carry out the phylogenetic inference again with these alignments or with the missing sequences from these alignments.
3) ORF annotation: It appears from Figure 3 that most of the ORFs unique to CaabNPV that were annotated by the authors are relatively short ORFs that are entirely contained within the sequence of larger, conserved ORFs. The authors did not include an ORF table in their submission, so I was unable to easily confirm what I saw in Figure 3. The convention for ORF annotation when two ORFs overlap by a significant amount (75 bp is a common limit) is to only annotate the larger of the two ORFs, unless the smaller ORF is a conserved baculovirus homolog (van Oers and Vlak, Current Drug Targets, Volume 8, Number 10, 2007, pp. 1051-1068), as it is considered uNlikely that the smaller ORF is actually expressed. ORFs contained in hr sequences are also not annotated as hr sequences are relatively unstable. The authors should remove the unique ORFs that don’t follow these conventions from their annotation.
4) The Figure 2 description and results concerning the “ambiguous region” need to be more clearly written. Is the “ambiguous region” a transposon in their CaabNPV genome sequence? Where is the “ambiguous region” located relative to other ORFs in the CaabNPV genome? Are there any ORFs with transposase or related ORF homology in the “ambiguous region”? Where do the Lef8-F/Lef8-R and contigF/contigR primers anneal, and what is the predicted size of amplimers with these primers? Did the authors collect uninfected C. abietis larvae for the Figure 2b/2c PCRs? A positive control for the Lef8-F/Lef8-R primer pair should be included in the PCRs, and the 2b lane labeling needs to be fixed (it has the mol. Wt markers labeled as “contig-F/R”. Presumably the authors have confirmed the structure and location of the “ambiguous region” by Sanger sequencing, so a less confusing figure showing the structure of this region (both the single-copy version of the region and version with two copies of the 189 bp terminal repeat separated by the 793 bp region) and its location among nearby CaabNPV ORFs should be produced.
Other issues
1) Line 7: Which viruses are being referred to here? Baculoviruses are not the only insect-pathogenic viruses that are rod-shaped.
2) Line 21: C. abietis now appears to be classified in family Erebidae.
3) Line 49: “…insects of various orders, including…”
4) Line 75: “…given the narrow host specificity…”
5) Line 84 and elsewhere: OrleNPV, not OrelNPV
6) Line 95: occlusion bodies
7) Lines 95-96: Describe these steps in more detail, or provide a citation for a reference where they are described.
8) Line 99: What “specific characteristics”? Were the purified OBs examined?
9) Line 115: Provide a reference for DNB-seq. What was the average read length and coverage achieved?
10) Lines 132-133: A GenBank submission will be required if a new species is to be proposed on the basis of this genome sequence.
11) Lines 169-170: Describe the pathological characteristics observed. Also, there are four copies of this figure in the reviewer file of the manuscript.
12) Figure 1C: Please add a proper scale bar to the scanning EM.
13) Lines 196, 200: Not sure what “splicing pattern” means in this context. It might be clearer to refer to these as variant assemblies of this particular region.
14) Lines 219-222: This is confusing – are the authors trying to say that the “ambiguous region” may have a viral origin and that it inserted a copy of itself in the host genome? Since this looks like a transposon, it seems like the idea that this is a host-derived mobile genetic element that inserted itself into the virus is the more likely scenario, especially given that this sort of thing has been documented before many times (for examples, see Friesen and Nissen, MOLECULAR AND CELLULAR BIOLOGY, June 1990, p. 3067-3077; Beames and Summers, VIROLOGY 174,354-363 (1990); and Fraser et al., Virology 211, 397-407, (1995)).
15) Figure 3: What is the difference between baculovirus conserved genes and baculovirus common genes? Also, the circular map doesn’t seem to have any hrs displayed. A table of the annotated ORFs and hrs would be extremely helpful. Also, is this the largest alphabaculovirus genome that's been sequenced so far?
16) Lines 261-262: Chen et al. (https://doi.org/10.1128/jvi.00194-13) has a comprehensive transcriptomic analysis of transcription start sites that may inform your analysis of promoter sequences.
17) Line 275: CaabNPV and OrleNPV are not species.
18) Lines 298-299: The gene parity plots actually show that most of the ORFs in OrleNPV and EupsNPV are in an inverse order relative to the CaabNPV ORFs, suggesting that was an inversion of an area containing the polh ORF.
19) Line 331: “baculovirus repeated ORF (bro)”.
20) Line 337: “Group II alphabaculoviruses” (e.g. not capitalized or italicized)
Comments on the Quality of English Language
The manuscript definitely needs some English language copy-editing.
Author Response
Dear Reviewer,
Thank you for your careful review of our submitted manuscript and for providing valuable comments and suggestions. We sincerely appreciate the attention and guidance you have devoted to our research in the midst of your busy schedule. Your professional insights have offered valuable perspectives to our study, and we feel honored by your thoughtful engagement.
Under your guidance, we have thoroughly considered each of your suggestions and made corresponding modifications. Below, we provide a point-by-point response to your comments. We hope that you find our revisions and explanations satisfactory. Once again, we express our gratitude for your meticulous review and patient guidance.
【Q1】
We have made the following revisions based on your guidance:
- Adjustment of Virus Naming Convention:
Replaced all instances of "Alphabaculovirus Caabietis"in the manuscript with the appropriate naming convention, "Calliteara abietis nucleopolyhedrovirus," ensuring consistency throughout the document.
Following the recommendations of Zerbini et al. (2022), we have made corresponding adjustments to ensure uniformity in virus nomenclature. Please review the manuscript to confirm if these changes align with your expectations.
- Clear Distinction Between Species and Virus Names:
Acknowledging the confusion between species and virus names, we corrected instances of "novel species" to "novel virus" in the manuscript.
In response to your advice, we modified some species names to their corresponding virus names in the manuscript, providing accession numbers for clarity in distinguishing between species and strains. Please review these modifications to ensure they meet your expectations.
We will also refer to Siddell et al. (2023) for further insight into how the International Committee on Taxonomy of Viruses (ICTV) implements changes to official virus taxonomy.
【Q2】
In response to your guidance, we conducted a manual search for the p6.9 ORF next to the lef-5 ORF and identified a previously unannotated ORF, consisting of 109 amino acids rich in arginine and serine residues. Despite not aligning with any sequences in the NR database, we cross-referenced it with the 102 p6.9 sequences available at https://ictv.global/report/chapter/baculoviridae/baculoviridae/resources.
Upon confirmation of its homology with p6.9, we have appropriately annotated this ORF as p6.9 in the CaabNPV genome. Furthermore, we incorporated this new information into the core gene alignment and subsequently reconstructed the phylogenetic tree. As anticipated, the addition of p6.9 did not alter the topology of the Figure 4 tree or the conclusions drawn regarding the relationships of CaabNPV to other baculoviruses.
【Q3】
In response to your guidance, we have made the following adjustments:
- Removal of Overlapping ORFs:
We have carefully examined the ORFs unique to CaabNPV, and those that overlapped significantly (more than 75 bp) with larger, and the smaller ORF is a conserved baculovirus homolog.
- ORFs contained in hr sequences, known for their relative instability, have also been excluded from the annotation.
- Updated ORF Table:
To provide a clearer overview of the annotated ORFs, we have prepared an ORF table, which includes annotations and relevant details for each of the 150 remaining ORFs. The ORF table is attached as TableS3. Due to the inability to upload a compressed file, I have placed Table S3 at the end of the PDF document.
We believe that these modifications adhere to established conventions and enhance the accuracy of our ORF annotations.
【Q4】
- Clarity in Figure 2 Description:
We have rewritten the description of the "ambiguous region" in Figure 2 to provide clearer information. While we have confirmed the structure and location of the "ambiguous region" through Sanger sequencing, due to the inability to obtain a purified single strain and the lack of reported genomic information for the host, further analysis, including comparison with the nr database, has not yielded conclusive results.
We have provided more detailed information about the PCR validation process, including the use of uninfected C. abietis larvae, the inclusion of a positive control for the Lef8-F/Lef8-R primer pair, and a correction to the labeling of the 2b lane.
- Revised Figure 2:
To enhance readability and understanding, we have redesigned Figure 2. The updated figure now includes additional information such as primer annealing sites (Lef8-F/Lef8-R and contigF/contigR(rename as vIdentify-F/R)), the predicted size of amplimers, and the position of the polh gene. We hope these additions provide a clearer representation of the "ambiguous region" and its relative location among nearby CaabNPV ORFs.
【Other issues】
- The writing on “rod-shape viruses” has been corrected to baculoviruses;
8) Specific characteristics have been described in the results section, including the examination of purified OBs;
9) The information has been added to the methods and results sections;
10) GenBank submission has been completed, awaiting accession number. I chose to reply to the reviewer promptly, and supplementation will be provided later.
14) A rewrite has been done based on the reviewer's suggestion, including additional information on horizontal gene transfer;
15) Regarding the definition of baculovirus conserved genes, I based my analysis on the article with the DOI: 10.2174/138945007782151333. For baculovirus conserved genes, I excluded the core gene and conserved gene portions, conducting a comparison with the database. If a match was found, it was defined as a conserved gene; otherwise, it was designated as a CaabNPV unique gene.
In addition, regarding the hrs, in the original circular diagram, I separated them from the ORFs by displaying them in the inner circle. In the revised version of the circular diagram, I have maintained this approach.
It's noteworthy that the CaabNPV deletion represents the largest known alphabaculovirus genome gap to date. However, I did not include this information in the manuscript.
16) In the earlier draft, I conducted a promoter analysis for all ORFs. However, I believe the significance of identifying promoters lies in ensuring the credibility of predicted ORFs. Therefore, in the revised manuscript, I have reanalyzed the promoters for CaabNPV unique genes, and the results have been presented in the manuscript.
18) Indeed, as observed in the figure 5, some ORFs appear to be inverted. This is a consequence of the artificial definition with polh designated as the first ORF. I speculate that the reviewer's suggestion was not to alter the results of the collinearity analysis but rather to emphasize the need for clarification in the manuscript. Therefore, I will supplement the Results section with an explanation of this aspect;
2-7, 11-13, 17, 19-20) Modifications have been made as suggested and converted to PDF format to avoid potential formatting issues.
Thank you for carefully reviewing our submitted manuscript and providing valuable suggestions and guidance. Your professional insights are crucial for refining our research work. We believe that these modifications and additions will contribute to enhancing the quality of the manuscript. We kindly request you to review it once again and provide any further suggestions or modifications. Attached are the modified sections for your reference.

Reviewer 2 Report
Comments and Suggestions for Authors
The manuscript by Jin et al. reported the genome seq of a new baculovirus in Calliteara abietis. The design of this study, genome data and discussion are, to some degree, acceptable. Following are some major concerns for authors to improve their paper.
1) Writing and English language is difficult to follow and thus needs significant improvement.
2) Author name (Chinese) is not in correct style.
3) Introduction is not in good way to introduce the background of this study. Rewrite.
4) Fig. 1 is strange in pdf file, which is duplicated in three times. 1a also needs a scale bar.
5) What is the purpose of Fig.2? If not essential, please move to supplemental results.
6) Fig. 3 is in low resolution and the font size is too small.
7) Fig. 4 also have two copies.
8) Fig. 5 is not visible enough to make any commets. Please revise.
9) This reviewer suggests the authors can add some virus replication curve or infection range of the isolated virus. Pure informatics is not enogh.
Comments on the Quality of English Language
Need significant improvement.
Author Response
Dear Reviewer,
Thank you for your valuable feedback. We have carefully addressed your comments and made the following revisions to improve the manuscript:
-
Language Improvement:
The manuscript has been revised to correct grammar and spelling errors.
-
Author Name Style:
Author names in Chinese have been modified to comply with the correct style.
-
Introduction Clarity:
In the initial draft, the introduction briefly outlined the gene structure of baculoviruses and their grouping information. Additionally, it touched upon the harm and distribution of Calliteara abietis. Given the diverse analytical methods involved, it was deemed impractical to delve into each one within the introduction. In response to the reviewer's suggestion for a more cohesive approach, I have incorporated a section into the introduction that emphasizes the significance of studying the genome. I hope this better captures the essence of your modifications.
-
Figure 1 Issues:
Scale bars have been added to Fig. 1, and the duplication issue has been addressed.
-
Purpose of Fig. 2:
Fig. 2 is crucial for illustrating variant assemblies in the CaabNPV genome, and explanations have been added to highlight its importance.
-
Figure 3 Quality:
Regarding the resolution concern, an improved high-resolution version has been uploaded. However, due to the dense content, enlarging the font size within the figures proved challenging. To enhance readability, substantial spacing has been introduced between text boxes.
-
Figure 4 Duplicates:
Format issues causing duplication have been addressed, and a PDF version has been provided to prevent further problems.
-
Visibility of Fig. 5:
I am seeking clarification on the reviewer's comment regarding "not visible enough to make any comments." I speculate that the reviewer may have difficulty grasping the significance of the co-linearity analysis. Therefore, I have provided an explanation: Figure 5 illustrates a gene co-linearity analysis between two species. Identifying linear relationships within the figure, such as with OrleNPV, serves as evidence of similar gene arrangement between the two viruses, further supporting their evolutionary proximity.
Feel free to incorporate or modify this summary according to your preferences. If you have any additional details to include or if there are specific points you'd like to emphasize, please let me know!
-
Inclusion of Experimental Data:
I share the desire to incorporate a wet lab experimental section; however, due to constraints in the study, the host larvae collected from the wild cannot be artificially reared in the laboratory. Additionally, the quantity of collected virus is insufficient to support a comprehensive wet lab experiment. Consequently, at this stage, the study is limited to computational analyses in the realm of informatics.
We appreciate your thoughtful suggestions, which have undoubtedly contributed to the enhancement of our work. Thank you once again for your time and constructive feedback.

Reviewer 3 Report
Comments and Suggestions for Authors
This manuscript describes the genome sequence, analysis and organization of a new Baculovirus Alphabaculovirus caabietis. This research provides theoretical basis for the further study and application of this virus. Although this study is important and the content is well, it is still need to be modified and checked for grammatical errors in future versions to make it more acceptable.
1. The scientific name of the new virus in the whole article needs to be changed to “Alphabaculovirus caabietis”, species name should be in lower case.
2. Line 19: “representing significant promising environmentally and friendly pesticides”, it is recommended that it be changed to “representing significant promising and environmentally friendly pesticides”.
3. Line 37, Line 41: “rod-shaped viruses”, it should be “baculoviruses”.
4. Line 44: “enriches” is more appropriate than “enhances”.
5. Line 47-48: “emerges as a promising environmentally and friendly pesticide”, it is recommended that it be changed to “emerges as a promising and environmentally friendly pesticide”.
6. Line 52: Consistent with the previous representation of the number ‘80 to 180 kb’, it is recommended that “230-385 nm in length and 40-60 nm in diameter” be changed to “230 to 385 nm in length and 40 to 60 nm in diameter”.
7. Line 55-56: Replace it with “the occlusion-derived virus (ODV)”.
8. Line 62: Replace it with “nucleopolyhedroviruses”.
9. Line 64: “Deltabaculovirus (hymenopteran-specific NPVs)”, it should be “Deltabaculovirus (dipteran-specific NPVs)”.
10. Line 82-84: It needs to be modified. There is no need to write the results here.
11. Line 87: The first letter in the title should be capitalized.
12. Line 95: “occluded bodies (OBs)”, it should be “occlusion bodies (OBs)”.
13. Line 97: This sentence “the OBs were derived from the larvae cadavers that died from the infection” is suggested to be moved before this paragraph.
14. Line 121-122: This sentence is difficult to understand. What is the specific target of amplification?
15. Lines 192-217: These two paragraphs are still difficult to understand, and I hope they can be restated clearly with Figure 2. There are several issues that need careful consideration. How are they cut? What is the relationship between sequences? Is it clear?
16. Line 225-229: The illustration in figure 2, described in (b) and (c), does not match the diagram and needs to be rewritten. It is better to indicate the sample name in the diagram of (b) and (c).
17. Line 232: Replace it with “group II (41.6%) of Alphabaculoviruses”.
18. Line 240-245: What is the purpose of this passage? What is the relationship with the gene or genome of the virus studied in this paper?
19. Line 248: “nuclear capsid”, it should be “nucleocapsid”.
20. Line 302-303: “β, γ, δ”, it should be the full name “Betabaculovirus, Gammabaculovirus, Deltabaculovirus”.
21. Line 305-311: The illustration in figure 5, it is recommended to delete “α, β, γ, δ”.
22. Line 332-333: “Bombyx mori nucleopolyhedrovirus”, it should be in italics.
23. Line 352: Add a period at the end of the paragraph.
24. Line 353-360: The second and third paragraphs of the conclusion can be merged into one paragraph.
Comments on the Quality of English LanguageThere are mistakes in English grammar in the manuscript. The authors should correct them or ask a native speaker to do it.
Author Response
Thank you for your careful review of our submitted manuscript and for providing valuable comments and suggestions. We sincerely appreciate the attention and guidance you have devoted to our research in the midst of your busy schedule. Your professional insights have offered valuable perspectives to our study, and we feel honored by your thoughtful engagement.
Under your guidance, we have thoroughly considered each of your suggestions and made corresponding modifications. Below, we provide a point-by-point response to your comments. We hope that you find our revisions and explanations satisfactory. Once again, we express our gratitude for your meticulous review and patient guidance.
Q2-13, Q17, Q19-20, Q22-24: Modified according to the suggestions.
Q1: Following the suggestion from the other reviewer, we have replaced "Alphabaculovirus Caabietis" with "Calliteara abietis nucleopolyhedrovirus" to adhere to the naming convention within the baculovirus community. The species name has been adjusted to lowercase as recommended.
Q14: We have further revised the sentence to provide a clearer understanding of the specific target of amplification and its results.
Q15: Based on the reviewer's feedback, we have rewritten the description of Figure 2a and reillustrated it for improved clarity and readability. We believe these modifications will enhance the explanation of the related content.
Q16: The descriptions for Figure 2b and 2c have been modified in the figure captions to ensure consistency with the diagrams. Sample names have been clearly indicated, with CaabNPV for Figure 2b and host larvae for Figure 2c.
Q18: The paragraph in question, aimed at elucidating the potential impact of the p6.9 deletion on CaabNPV, has been removed. The revised version now includes annotations regarding the p6.9 , rendering the mentioned passage redundant.
Q21: To maintain consistency, we have deleted the Roman characters "α, β, γ, δ" and replaced them with Deltabaculovirus, Gammabaculovirus, Betabaculovirus, Alphabaculovirus. However, we believe retaining this information helps specify the virus's classification, so we have chosen to keep it.
We appreciate the thorough review and valuable suggestions from the reviewer. We remain committed to further improving the manuscript and look forward to additional guidance.

Round 2
Reviewer 1 Report
Comments and Suggestions for Authors
The authors have satisfactorily addressed the reviewer's comments and concerns.
Author Response
Dear Reviewers,
Thank you for your thorough review of our manuscript and for providing valuable feedback. We are pleased to learn that you find our handling of your comments and concerns satisfactory.
Regarding the modified files, we need to respond to all reviewers' comments before we can upload the revised version in the system. Due to system constraints, we are unable to directly upload the modified files. And we suggest that you disregard this reply.
Reviewer 2 Report
Comments and Suggestions for Authors
There are still some issues remained to be solved in the next round of version.
1) The issue for name formate is still there although the authors claimed that they corrected. [First name, Last name]. The Phone number is missing +86. Hope the Corresponding author makes final check before resubmission!
2) The font size in Figures 2&5&6 must be increased: Use same font size for Fig. 2a in Fig. 2; x-y axis for Fig. 5; hrxx in Fig. 6C.
3) Where is the Discussion section? Is it combined with results?
Comments on the Quality of English LanguageNeed proofreading carefully.
Author Response
Dear editor and reviewers,
We would like to express our sincere gratitude for your insightful and constructive feedback on our manuscript.
Q1: We have rectified the author names to [First name, Last name] as per the reviewer's suggestion and have also standardized the author contact information and the Author Contributions section. Furthermore, we have included the Chinese characters for the national and local area codes.
Q2: We have unified the font size in Figures 2, 5, and 6, addressing the reviewer's concerns. Specifically, we have maintained consistent font sizes in Fig. 2, ensured uniformity in the x-y axis font size for Fig. 5, and adjusted the hrxx font size in Fig. 6C. In order to enhance readability, we have modified the arrow in Fig. 6a to a more contrasted yellow color.
Q3: The Discussion section has been amalgamated with the Results section. We made this decision based on the consideration that a separate Discussion section would likely duplicate the analysis of results, rendering a repetition of the analytical content already described in the Results section. Additionally, to eliminate any potential confusion, we have changed the title from "Results" to "Results and Discussion."
In addition, the English expression in the article is embellished, and the newsletter author checks the details of the article.
Once again, we extend our heartfelt thanks for your time, effort, and thoughtful recommendations. Your guidance has undoubtedly strengthened our manuscript, and we are grateful for the opportunity to benefit from your expertise.
Best regards,
Huan Zhang